# Prenatal Noninvasive Trio-WES in a Case of Pregnancy-Related Liver Disorder

**DOI:** 10.3390/diagnostics11101904

**Published:** 2021-10-14

**Authors:** Aldesia Provenzano, Antonio Farina, Anna Seidenari, Francesco Azzaroli, Carla Serra, Anna Della Gatta, Orsetta Zuffardi, Sabrina Rita Giglio

**Affiliations:** 1Medical Genetics Unit, Department of Experimental and Clinical Biomedical Sciences “Mario Serio”, University of Florence, 50139 Florence, Italy; 2Division of Obstetrics and Prenatal Medicine, IRCCS Sant’Orsola-Malpighi Hospital, University of Bologna, 40138 Bologna, Italy; antonio.farina@unibo.it (A.F.); anna.seidenari@gmail.com (A.S.); anna.ndg1@gmail.com (A.D.G.); 3Division of Internal Medicine, IRCCS, Azienda Ospedaliero-Universitaria di Bologna, 40138 Bologna, Italy; francesco.azzaroli@unibo.it; 4Department of Organ Failure and Transplantation, Sant’Orsola-Malpighi Hospital, Via Massarenti 9, 40138 Bologna, Italy; carla.serra@aosp.bo.it; 5Medical Genetics Unit, Department of Molecular Medicine, University of Pavia, 27100 Pavia, Italy; orsetta.zuffardi@unipv.it; 6Medical Genetics Unit, Department of Medical Sciences and Public Health, University of Cagliari, 09126 Cagliari, Italy; sabrinar.giglio@unica.it

**Keywords:** fetal cell free DNA, exome sequencing, non-invasive WES, liver disorder

## Abstract

Liver disease in pregnancy may present as an acute condition related to the gestational period, characterized by pruritus, jaundice, and abnormal liver function. The disease may be misdiagnosed with other liver diseases, some of which may have consequences for fetal health. It is therefore advisable to implement rapid diagnostic strategies to provide information for the management of pregnancy in these conditions. We report the case of a healthy woman with a twin pregnancy from homologous in vitro fertilization (IVF), who in the third trimester presented jaundice and malaise. Biochemical investigations and liver hyperechogenicity raised the suspicion of acute fatty liver disease of pregnancy (AFLP). Non-invasive prenatal whole-exome sequencing (WES) in the trio identified the Phe305Ile heterozygous variant in the *ATP8B1* gene. Considering the twin pregnancy, the percentage of the variant versus the wild allele was of 31%, suggesting heterozygosity present in the mother alone. This analysis showed that the mother was affected by benign recurrent intrahepatic cholestasis of pregnancy (ICP1: # 147480) and indicated the opportunity to anticipate childbirth to avoid worsening of the mother’s health. WES after the birth of the twins confirmed the molecular data.

## 1. Introduction

Numerous liver diseases can occur during pregnancy, and some of them can be accompanied by serious consequences for the mother and the baby. They often have very similar presentation, and their diagnosis does not seem to be obvious in all cases [1,2,3]. Pregnancy-related liver disorders generally occur in the second or third trimester, with a mortality up to 25% [4].

Differential diagnosis between hyperemesis gravidarum (HG), intrahepatic gestational cholestasis (ICP), preeclampsia (PE), acute fatty liver of pregnancy (AFLP), and/or HELLP syndrome is necessary to ensure the diagnostic outcome and the exact interdisciplinary management of the pregnancy [5,6,7,8].

A clinical and biochemical investigation does not fully discriminate between the various forms [9,10,11]. In particular, AFLP and ICP1 share most clinical signs and occur in the same period of pregnancy, so that DNA analysis becomes imperative for a precise classification of the disease and the consequent therapeutic decisions. ICP1 causes severe itching that usually begins on palms and soles and then spreads to other parts of the body. Although the condition is often not strictly severe, it can cause problems for the fetus. In fact, it is associated with an increased risk of premature birth and stillbirth [11]. Additionally, some babies born to ICP1 mothers present an increased risk of adverse perinatal outcomes [12].

We describe the case of a woman with diamniotic dichorionic twin pregnancy (DCDA) who arrived at clinical observation at 32 + 3 weeks of pregnancy for generalized itching, evocative of intrahepatic cholestasis, with liver enlargement suggestive of severe steatosis. This result, together with leukocytosis and increased serum transaminases and uric acid led to the suspicion of AFLP, five Swansea criteria being met. Considering the importance of obtaining an accurate diagnosis to improve the clinical management of both the mother and the fetuses and to cope with subsequent pregnancies in the best possible way, we decided to elucidate the basis of the disease through DNA investigation also by a noninvasive procedure.

In fact, since variants of some genes associated with AFLP can result in severe disease of the newborn if at homozygous or compound heterozygous state, we decided to analyze at the same time both fetal and parents’ DNA.

## 2. Materials and Methods

### 2.1. Patient

The study conformed to the guidelines outlined by the 1975 Declaration of Helsinki, and written consent was obtained from the patient. A 35-year-old female primigravida at 32 + 3 weeks of a dichorionic diamniotic (DCDA) twin pregnancy from homologous in vitro fertilization (IVF) was admitted to the division of prenatal medicine of Sant’Orsola-Malpighi Hospital. In line with the IVF protocol, the patient received combined estrogen and progesterone supplementation until gestation week 12. The woman had a BMI of 33 kg/m^2^ at the appearance of the symptoms (having gained 12 kg in weight). At the time of admission, due to the sudden appearance of abdominal pain as well as generalized itching, laboratory tests showed a mild increase in transaminases above the upper limit of normal and elevated serum bile acid concentrations (23 μmol/L), a picture consistent with ICP; therefore, treatment with ursodeoxycholic acid 1500 mg/day was started. Abdominal ultrasonography revealed an enlarged liver (longitudinal right lobe diameter, 20 cm) with regular margins and marked hyperechogenicity determining attenuation of the ultrasound beam in the deeper segments of the liver, suggestive of severe steatosis (Figure 1); due to the ultrasound beam attenuation, the suprahepatic veins were not visible. Liver elastography with 2Dlogic E9 gave a liver stiffness value of 1.48 ± 0.13 m/s.

During her stay in the ward, neutrophil leukocytosis (>15,000 cells/μL), increased serum transaminases (up to a 3-fold upper limit of normal), and increased serum uric acid (399 μmol/L) were observed. In addition to worsening transaminases, hyperuricemia, leukocytosis, nausea, abdominal pain, and a marked bright liver on ultrasound, the patient fulfilled five Swansea criteria for AFLP [13].

Given the possible fetal involvement in AFLP, we opted for WES analysis of the blood DNA of the woman and her partner and of the cell-free fetal DNA (cffDNA) of the ongoing pregnancy to search for fetal variants. The parents signed informed written consent prior to sample collection.

WES analysis of the blood DNA of the woman and her partner and of the cell-free fetal DNA (cffDNA) of the ongoing pregnancy led to a diagnosis of ICP1.

During hospitalization, the patient exhibited worsening serum bilirubin (from 0.7 to 17 μmol/L) and therefore underwent a cesarean delivery at 35 weeks after receiving antenatal betamethasone to accelerate fetal lung maturation. After delivery and then every six months, the patient was followed up. The last known follow-up, carried out 20 months after pregnancy, showed normal liver function, a residual hyperechoic area of 16 × 10 mm at liver segment VI (also investigated by contrast US imaging and suggestive of an area of residual steatosis), and normal liver dimensions.

After birth, twins auxological parameters and Apgar scores were appropriate. Clinical examination at 2 and 4 months of age confirmed normal growth and psychomotor development.

### 2.2. DNA Extraction, Library Construction, and Sequencing

Cell-free fetal DNA (cffDNA) and genomic DNA (gDNA) extraction and sequencing were performed as previously described [14]

### 2.3. Estimation of the cffDNA Fraction

To determine the cffDNA fraction, the proportion of reads on the Y chromosome, in particular, the presence of the variants in the SRY gene, was estimated. Taking into consideration that in twin pregnancies the cffDNA fraction ranges from 5.4% to 23.5% [15], the distribution of the cffDNA fraction in twins has an average of 12.1% as compared to 9.6% in individual pregnancies in our in-house samples. The presence of the Y chromosome and 32 non polymorphic Y-linked loci indicated that the twins were dizygotic males. After normalization of the frequency of the X and the Y chromosomes combined with the cffDNA fraction and the analysis of autosomal single-nucleotide polymorphisms (SNPs) having a frequency greater than 2%, we confirmed the dizygosity of the male twins, as expected by IVF, and estimated the total cffDNA as approximately 10%.

### 2.4. Data Analysis

WES analysis was performed by examining the filtered FastQ data of all genes hitherto known to be associated with pregnancy and liver disease and following manual IGV screening of BAM files.

Variant calling was carried out using Varscan software (version 2.4.0, Washington University, Greater St. Louis, MO, USA) to call variants which met the desired thresholds for read depth, base quality, variant allele frequency, and statistical significance. This software allows the analysis of the variants present in regions with low coverage, somatic mutations and multisamples, as well as germline variants. This strategy was applied to both cffDNA and genomic WES.

## 3. Results

WES was performed in 2 weeks, while the confirmation of the DNA analysis was carried out a few days after the birth of the twins.

Trio-WES (cffDNA plus maternal and paternal gDNA) identified the c.913T > A variant (Phe305Ile) in the *ATP8B1* gene in maternal cffDNA and lymphocytes DNA. No variants were detected along other liver disease-associated genes. Phe305Ile (rs150860808) is reported in the gnomAD database (v.2.1.1) with a frequency of 0.18% in the non-Finnish European population (NFE) (0.19% in males and 0.16% in females), and no homozygotes are reported, while in general populations, the frequency is 0.12% and 0.1% in males and females, respectively. In the ClinVar database, the variant is reported with conflicting interpretations of pathogenicity.

The cffDNA was analyzed at a greater reading depth than the maternal and paternal DNA (Figure 2), and the lymphocytes’ DNA of the twins was also analyzed after birth. Since in cffDNA the fetal fraction was ~10% and the variant reading rate was ~31%, a clear imbalance towards the wild-type allele emerged (Figure 2), suggesting that the variant was exclusively maternal. All lymphocytes DNA data confirmed the interpretation.

The repeated bile acid analysis in the mother 10 after delivery of the twins was just above the normal values.

## 4. Discussion

Disease framing through DNA analysis can significantly increase the diagnostic rate and lead to better management, regardless of the specificity of the clinical signs. This takes on effective significance in the prenatal setting, where an exact diagnosis can greatly influence the management of both pregnancy and the newborn.

AFLP is a rare disease of the third trimester of pregnancy (1 in 7000–10,000 pregnancies), associated with variants of the *LCHAD* gene alone, while ICP, a disorder of the second or third trimester of pregnancy, is found in 3–5% of pregnancies, with variants associated with 13 genes of the ABC membrane protein complex, and ATP8B1, TJB2, NR1H4, and ANO8 involved in the transport and metabolism of bile salts. On a clinical basis, these pregnancy disorders are more severe in AFLP, with the need for immediate termination of the pregnancy to avoid maternal disseminated intravascular coagulation and death. If the newborn is a carrier of the mother’s genetic defect, the risk of death from dilated cardiomyopathy or progressive neuropathy is high. Instead, ICP can be treated with ursodeoxycholic acid (UDCA) to reduce the risk of perinatal morbidity and mortality and relieve maternal symptoms. The timing of delivery reflects the balance between fetal death, risk of severe prematurity, and sudden maternal death with high concentrations of bile acids. In the absence of genetic tests, the differential diagnosis between AFLP and ICP can be challenging, and the analysis of fetal and maternal DNA may be appropriate to identify any risks to the fetus, if it is a carrier of the maternal genetic variant.

In our case, WES identified a maternal heterozygous variant (c.913T > A) located in exon 10 of the *ATP8B1* gene (NM_005603.6), leading to diagnosis of ICP1, anticipation of the delivery, and setting up of a follow-up program for liver function checking. ATP8B1 is a flippase involved in phospholipid translocation from the outer to the inner leaflet of the cell membrane, expressed in the canalicular membrane of liver cells as well as in the apical membranes of intestinal and pancreatic cells [16,17]. The variant we detected has previously been reported in individuals with ICP1, all showing post-partum resolution [18]. Thus, the variant, although predicted as possibly deleterious, does not seem to have effects other than impaired liver function during pregnancy, at least at the heterozygous state. Indeed, combined heterozygosity for this variant as well as for other variants of *ATP8B1* is associated with progressive familial intrahepatic cholestasis (#243300), characterized by episodic cholestatic disease with intervals during which there is no clinical or biochemical evidence of cholestasis. This indicates that intrahepatic cholestasis linked to this gene is triggered not only by variants that alter the stability of the protein, its folding, or the ability to interact with other proteins involved in the transport of bile salts, but also by the environment. During pregnancy, the higher estrogen levels can impair biliary salts transport to the liver, determining cholestasis in predisposed individuals [19]. It is possible that even an excessive increase in body weight, as was the case for our proband, can increment the predisposition to cholestasis. As to intermittent cholestasis associated to *ATP8B1*, bacterial endotoxins such as lipopolysaccharides released from sites of infection [20] might induce the failure of bile transport, thus explaining intermittent cholestasis. Therefore, drugs that act on the transport of bile salts may prevent the occurrence of these conditions.

Our study demonstrates that (i) clinical examination alone is not sufficient for a correct diagnosis, even in the presence of specific clinical/biochemical signs; (ii) diagnosis by noninvasive procedures can be highly efficient even for twin pregnancies; (iii) WES on cffDNA and parental genomic DNA can provide reliable results in few weeks and at a cost comparable to that of SN or CGH-array, even if deeper sequencing is required for cffDNA; (iv) genetic data strongly help in the management of the patient and, in prenatal diagnosis, even of the unborn child.

Finally, this study confirms that the effect on health and disease of frequent variants depends on the burden of environmental factors [21].

## 5. Take Home Message

WES may help the diagnosis of conditions where the phenotypes are not immediately interpretable. This assumes an effective significance in prenatal diagnosis, since a correct diagnosis can notably influence the management of the patient in terms of drug therapy and the management of the newborn.

Liver disease can occur during pregnancy where signs and symptoms can sometimes overlap, and usually its features become evident in the third trimester of pregnancy. Since these are complex conditions that often share clinical manifestations, it is essential to find a method for a rapid and precise diagnosis in order to distinguish between the different liver diseases.

## Figures and Tables

**Figure 1 diagnostics-11-01904-f001:**
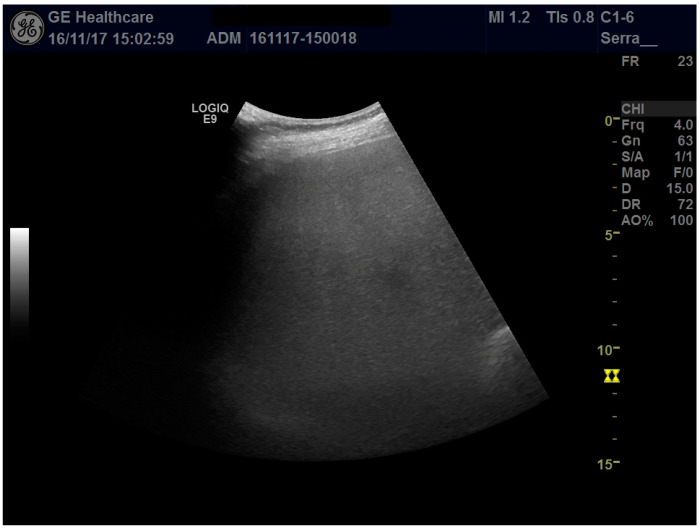
Ultrasound examination of the right liver lobe: the enlarged liver appears typically as “bright”, with fine, closely packed echoes without visualization of the vessels, as in the case of severe hepatic steatosis.

**Figure 2 diagnostics-11-01904-f002:**
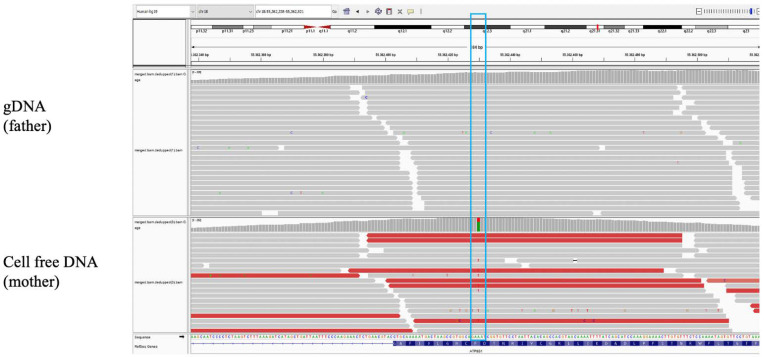
Integrative Genomics View (IGV) representation of the *ATP8B1* variant in cffDNA and in paternal gDNA. The *ATP8B1* missense variant c.[913T > A] + [=]p.[Phe305Ile ] + [=], reported in the heterozygous state in dbSNP (rs150860808), gnomAD v2.1.1 (0.1%), and 1000G (0.09%), was detected in cffDNA with a ratio of 31% as compared to the wild-type allele. The imbalanced allelic ratio, wild-type (WT) allele/mutated allele, with an overrepresentation of the WT allele, would be associated with healthy fetuses. The WT allele is indicated in green, while the mutated allele is shown in red. Considering that the fetal fraction was 10%, these data suggested that the fetuses did not receive the missense maternal variant having a frequency of 31%, but rather those having a frequency of 50 or more readings. Postnatal WES confirmed that the two newborns were wild type for the *ATP8B1* variant. The cffDNA sequencing was carried out at a greater reading depth than the genomic one; cffDNA coverage average 277×, gDNA coverage average 103×. The blue box indicates the nucleotide change in the sequencing.

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
