# Peer review of "Prenatal Noninvasive Trio-WES in a Case of Pregnancy-Related Liver Disorder"

_diagnostics, 2021, doi:10.3390/diagnostics11101904_

Round 1

Reviewer 1 Report

The authors presented a case of liver disorder in pregnancy with differential diagnosis of intrahepatic cholestasis of pregnancy (ICP) and acute fatty of pregnancy (AFLP). By finding a variant in the ATP8B1 gene WES on maternal plasma cell-free DNA (cfDNA) and maternal gDNA, the authors made the diagnosis of ICP. This case is well written and can shed light on the genetic origin of ICP. However, using such genetic study to determine the diagnosis of a liver disorder in pregnancy and guide the subsequent management is not a standard method. In my opinions, the findings in this case are inadequate to support such approach, and there is still a knowledge gap. Revision of the discussion and conclusions is required before acceptance for publication.

Case

The authors wrote: ‘…on the cell-free fetal DNA (cffDNA) of the 88 ongoing pregnancy to search for fetal variants.’ Cell-free DNA or Cell-free fetal DNA? Would the authors clarify why did they want to search for fetal variants? Was the finding of fetal variants affect the diagnosis or management?

The authors wrote: ‘WES analysis on the blood DNA of the woman and her partner and on the cell-free fetal DNA (cffDNA) of the ongoing pregnancy led to a diagnosis of ICP1.’

Later, the authors wrote: ‘Since in cfDNA the fetal fraction was ~ 10% and the variant reading rate ~ 31%, a clear imbalance towards the wild type allele emerges (Figure 2), suggesting that the variant is exclusively maternal. The gDNA data confirms the interpretation.’

Would the authors clarify whether the diagnosis was made by gDNA analysis (second sentence) or cfDNA analysis (first sentence)?

Was WES performed with targeted gene mutations?

What was the turnover time of WES? Did the result came back before the twin birth ?

Was bile acid repeated during hospitalisation? If yes, what was the result?

Figure 1. The patient’s name should be removed. The image quality is suboptimal. The image focus was not in the near field. The echoes was too bright.  

Discussion

Would the authors discuss the prevalence of mutations associated with ICP, AFLP, and found in normal pregnant women? A recent study did not find any known mutations in AFLP.

The authors wrote: ‘…the genetic data strongly helps in the management of the patient and, in prenatal diagnosis, even of the unborn child.’ In my opinions, the genetic data may help making the diagnosis. Current management is based on clinical and biochemical markers, bile acids in particular. Would the authors discuss the principle of management of liver disorders with differential diagnosis of ICP and AFLP in light of the findings of this case? Did the authors mean the management could follow that of ICP when an associated gene variant was found even if Swansea criteria of AFLP was met? However, AFLP can progress rapidly and be potentially fatal. The turnaround time of WES is in weeks even it can improve the diagnosis.

The authors wrote: ‘…diagnosis by noninvasive procedure can be highly efficient even in twin pregnancies…’ Would the authors clarify the usefulness of performing gene mutations analysis in maternal plasma cFDNA while the diagnosis can be made by maternal gDNA? In this case, the main aim was intended to find whether the woman, but not the twin fetuses, had mutations.

Although the findings as presented suggest maternal contribution, whether fetal contribution was coexistent was not known.

The authors wrote: ‘estimated the total cffDNA as approximately 10%.’ Would the authors discuss their finding in comparison to a previous study which showed that cfDNA was higher and more fragmented in ICP?

The authors wrote: ‘WES can significantly increase the diagnostic rate where the phenotypes are not immediately interpretable.’ In view of the limitations as stated above, would the authors revise the text to ‘WES may help the diagnosis….’?  

Author Response

Reviewer 1:

The authors presented a case of liver disorder in pregnancy with differential diagnosis of intrahepatic cholestasis of pregnancy (ICP) and acute fatty of pregnancy (AFLP). By finding a variant in the ATP8B1 gene WES on maternal plasma cell-free DNA (cfDNA) and maternal gDNA, the authors made the diagnosis of ICP. This case is well written and can shed light on the genetic origin of ICP.

We thank the reviewer for the comment. The correlations between fetal pathology and pregnancy are still poorly understood. A robust example is maternal disomy of chromosome 16 and its obvious correlation with preeclampsia and premature birth. Similarly, variants of the LCHAD and CPT1A genes can be associated with maternal liver disorders in pregnancy if in the heterozygous state and with severe disease of the newborn or later if in the homozygous / compound heterozygous state (Lucian Gheorghe Pop et al, 2020, JIM, doi .org / 10.2478 / jim-2020-0001; Rachel H Westbrook et al, 2016 Journal of Hepatology, doi: doi.org/10.1016/j.jhep.2015.11.030).

To this end, in the introduction we added the sentence: “In fact, since variants of some genes associated with AFLP can result in severe disease of the newborn if at homozygous or compound heterozygous state, we decided to analyze both the fetal DNA and that of the parents at the same time”

The authors wrote: ‘…on the cell-free fetal DNA (cffDNA) of the 88 ongoing pregnancy to search for fetal variants.’ Cell-free DNA or Cell-free fetal DNA? Would the authors clarify why did they want to search for fetal variants? Was the finding of fetal variants affect the diagnosis or management?

We conducted the analysis on cell-free fetal DNA and not cell-free DNA. Accordingly, in the text we changed in cell-free fetal DNA.

As to the question ”Would the authors clarify why did they want to search for fetal variants?”, we have already clarified this point at the end of the introduction

The authors wrote: ‘WES analysis on the blood DNA of the woman and her partner and on the cell-free fetal DNA (cffDNA) of the ongoing pregnancy led to a diagnosis of ICP1.’

Later, the authors wrote: ‘Since in cfDNA the fetal fraction was ~ 10% and the variant reading rate ~ 31%, a clear imbalance towards the wild type allele emerges (Figure 2), suggesting that the variant is exclusively maternal. The gDNA data confirms the interpretation.’

Would the authors clarify whether the diagnosis was made by gDNA analysis (second sentence) or cfDNA analysis (first sentence)?

We tried to made more clear this point by re-writing the results and substituting gDNA with ”lymphocytes DNA”:

Trio-WES (cffDNA plus maternal and paternal DNA) identified the c.913T>A variant (Phe305Ile) in the ATP8B1 gene in maternal cfDNA and lymphocytes DNA. No variants were detected along other liver disease-associated genes. Phe305Ile (rs150860808) is reported in the gnomAD database (v.2.1.1) with

a frequency of 0.18% in the non-Finnish European population (NFE) (0.19% in males and 0.16% in females) and no homozygotes are reported, while in general populations the frequency is of 0.12% and 0.1% in males and females respectively. In the ClinVar database the variant is reported with conflicting interpretations of pathogenicity. The cffDNA was analyzed at a deeper reading depth than the maternal and paternal lymphocytes DNA (Fig. 2), and the lymphocytes DNA of the twins performed after birth. Since in cffDNA the fetal fraction was ~ 10% and the variant reading rate ~ 31%, a clear imbalance towards the wild type

allele emerges (Figure 2), suggesting that the variant is exclusively maternal. All lymphocytes DNA data confirm the interpretation.

Was WES performed with targeted gene mutations?

WES analysis was performed by examining both the filtered fastQ data of all genes hitherto known to be associated with pregnancy and liver disease and following manual IGV screening of BAM files

We clarified this point in the data analysis section

What was the turnover time of WES? Did the result came back before the twin birth ?

The WES was performed in 2 weeks, while the confirmation of the DNA analysis was carried out a few days after the birth of the twins.

Was bile acid repeated during hospitalisation? If yes, what was the result?

The repeated bile acid analysis in the mother 10 after delivery of the twins was just above normal values

Figure 1. The patient’s name should be removed. The image quality is suboptimal. The image focus was not in the near field. The echoes was too bright.  

Any possible reference to the patient's identity has been removed and image quality has been improved

Discussion

Would the authors discuss the prevalence of mutations associated with ICP, AFLP, and found in normal pregnant women? A recent study did not find any known mutations in AFLP.

The authors wrote: ‘…the genetic data strongly helps in the management of the patient and, in prenatal diagnosis, even of the unborn child.’ In my opinions, the genetic data may help making the diagnosis. Current management is based on clinical and biochemical markers, bile acids in particular. Would the authors discuss the principle of management of liver disorders with differential diagnosis of ICP and AFLP in light of the findings of this case? Did the authors mean the management could follow that of ICP when an associated gene variant was found even if Swansea criteria of AFLP was met? However, AFLP can progress rapidly and be potentially fatal. The turnaround time of WES is in weeks even it can improve the diagnosis.

The authors wrote: ‘…diagnosis by noninvasive procedure can be highly efficient even in twin pregnancies…’ Would the authors clarify the usefulness of performing gene mutations analysis in maternal plasma cFDNA while the diagnosis can be made by maternal gDNA? In this case, the main aim was intended to find whether the woman, but not the twin fetuses, had mutations.

Although the findings as presented suggest maternal contribution, whether fetal contribution was coexistent was not known.

We answered all these questions with this explanation that we added after the second sentence of the discussion

AFLP is a rare disease of the third trimester of pregnancy (1: 7000-10000 pregnancies), associated with variants of the LCHAD gene alone, while ICP, a disorder of the second or third trimester of pregnancy is found in 3-5% of pregnancies, with the variants associated with 13 genes of the ABC membrane protein complex, and ATP8B1, TJB2, NR1H4, and ANO8, involved in the transport and metabolism of bile salts. On a clinical basis, these pregnancy disorders are more severe in AFLP with the need for immediate termination of pregnancy, to avoid maternal disseminated intravascular coagulation and death. If the newborn is a carrier of the mother's genetic defect, the risk of death from dilated cardiomyopathy or progressive neuropathy is high. Instead, ICP can be treated with ursodeoxycholic acid (UDCA) to reduce the risk of perinatal morbidity and mortality and relieve maternal symptoms. The timing of delivery reflects the balance between fetal death, risk of severe prematurity and sudden maternal cardiac death in high concentrations of bile acidsIn the absence of genetic tests, the differential diagnosis between AFLP and ICP can be challenging and the analysis of fetal DNA as well as that of the mother may be appropriate to identify any risks to the fetus if it is a carrier of the maternal genetic variant

The authors wrote: ‘estimated the total cffDNA as approximately 10%.’ Would the authors discuss their finding in comparison to a previous study which showed that cfDNA was higher and more fragmented in ICP?

10% refers to fetal cfDNA present in maternal plasma. We have not performed cffDNA fragmentation studies because we have enough experience and controls to consider this approach reliable (see also: Provenzano A et al,  Prenat Diagn 2020, doi:101002/pd.5700)

The authors wrote: ‘WES can significantly increase the diagnostic rate where the phenotypes are not immediately interpretable.’ In view of the limitations as stated above, would the authors revise the text to ‘WES may help the diagnosis….’?  

We changed the text accordingly to your suggestion

Reviewer 2 Report

This is an extremely rare case report about the acute fatty liver of pregnancy (AFLP). I have some minor comments that could be taken into consideration for further improvement of the paper.

The abstract:

This section is appropriate.

Material and Methods:

Page 2, line 77: It would be easier to understand the article if you specify the normal range of liver stiffness values.

Discussion:

After giving birth, what was the baby's prognosis?

Do you have WES data tested from babies after birth?

In addition to ultrasonography, have you not performed other diagnostic tests for confirmation such as liver biopsy?

Author Response

Reviewer 2:

Comments and Suggestions for Authors

This is an extremely rare case report about the acute fatty liver of pregnancy (AFLP). I have some minor comments that could be taken into consideration for further improvement of the paper.

The abstract:

This section is appropriate.

Material and Methods:

Page 2, line 77: It would be easier to understand the article if you specify the normal range of liver stiffness values.

We added normal range values in the text

Discussion:

After giving birth, what was the baby's prognosis?

After birth, the newborns were in good health as stated at the end of the paragraph “patient”:

After birth, twins auxological parameters and Apgar scores were appropriate. Clinical examination at 2 and 4 months of age confirmed normal growth and psychomotor development.

Do you have WES data tested from babies after birth?

We performed WES analysis on peripheral blood after babies’ birth, as written in the results

In addition to ultrasonography, have you not performed other diagnostic tests for confirmation such as liver biopsy?

We did not perform other diagnostic tests because of the improved clinical parameters of the patient